# A Multidrug-Resistant *Escherichia coli* Caused the Death of the Chinese Soft-Shelled Turtle (*Pelodiscus sinensis*)

**DOI:** 10.3390/vetsci12050473

**Published:** 2025-05-14

**Authors:** Mingyang Xue, Xiaowei Hu, Nan Jiang, Wei Liu, Zidong Xiao, Chunjie Zhang, Yeying Wu, Tianwang Liang, Huixuan Zhang, Yuding Fan, Yan Meng, Yong Zhou

**Affiliations:** 1Yangtze River Fisheries Research Institute, Chinese Academy of Fishery Sciences, Wuhan 430223, China; xmy@yfi.ac.cn (M.X.); 13100852930@163.com (X.H.); jn851027@yfi.ac.cn (N.J.); liuwei@yfi.ac.cn (W.L.); xiaohzau@163.com (Z.X.); zcj1873199292@163.com (C.Z.); fanyd@yfi.ac.cn (Y.F.); 2Freshwater Fisheries Research Center, Chinese Academy of Fishery Sciences, Wuxi 214081, China; 3School of Life Science, Wuchang University of Technology, Wuhan 430223, China; colour0924@163.com (Y.W.); nntw1118@163.com (T.L.); 18171668310@163.com (H.Z.)

**Keywords:** *Pelodiscus sinensis*, *Escherichia coli*, multidrug-resistant, drug resistance genes

## Abstract

Disease outbreaks caused by bacterial pathogens, particularly antimicrobial-resistant strains, represent a critical challenge in Chinese soft-shelled turtle (*Pelodiscus sinensis*) aquaculture. In this investigation, researchers characterized a multidrug-resistant *Escherichia coli* strain (HD-593) isolated from moribund specimens. Antimicrobial susceptibility profiling revealed resistance to 14 clinically relevant antibiotics and had several resistance genes. Experimental challenge trials demonstrated significant pathogenicity, with a median lethal dose of 6.53 × 10^5^ CFU/g. Infected turtles had damaged organs and changes in serum levels. This study shows that this multidrug-resistant *E. coli* is a threat to aquaculture. It is important to use antibiotics carefully and find alternatives which can help protect animal and public health.

## 1. Introduction

The Chinese soft-shelled turtle (*Pelodiscus sinensis*), also known as *Trionyx sinensis*, is a valuable aquaculture species in East Asia, particularly in China, due to its high economic and nutritional value [1]. Production of *P. sinensis* has steadily increased in recent years, primarily in regions like Zhejiang, Anhui, Guangdong, Hubei, Jiangsu, and Jiangxi [2]. The industry has trended toward industrialization, standardization, and intensified production methods [3,4]. However, the increasing scale of *P. sinensis* aquaculture has been accompanied by the emergence of various diseases. Disease outbreaks, particularly bacterial infections, during the summer months pose a significant challenge for *P. sinensis* farmers. Several bacterial pathogens, including *Bacillus cereus* [5], *Aeromonas veronii* [6,7], *Aeromonas hydrophila* [3,8], and *Morganella morganii* [9], have been identified as causative agents of *P. sinensis* disease. Among the bacterial pathogens affecting aquaculture species, *Escherichia coli* is notable due to its dual role as a common gut bacterium and a potential pathogen under certain conditions. Although *E. coli* is generally non-pathogenic, certain strains have acquired virulence factors that enable them to cause infections in a variety of hosts, including aquatic animals [10].

Multidrug resistance among bacterial pathogens has become a major concern worldwide, driven by the widespread use and misuse of antibiotics in both human and animal medicine [11]. Treating infections caused by antimicrobial-resistant pathogens presents a significant challenge to global healthcare systems and the breeding industry, driving up costs and placing considerable strain on resources [12]. In aquaculture, the indiscriminate application of antibiotics to control bacterial infections has exacerbated this issue, leading to the emergence of antibiotic-resistant bacteria that can survive multiple classes of antibiotics. Such bacteria not only threaten the sustainability of aquaculture but also pose a potential risk for zoonotic transmission, creating a pathway for antibiotic resistance to spread to human populations.

This study documents a case of mortality in *P. sinensis* caused by a multidrug-resistant *E. coli* strain, aiming to provide insights into the pathogen’s resistance profile and pathogenicity. A strain, designated *E. coli* HD-593, was isolated from the diseased *P. sinensis*. Antibiotic susceptibility testing, combined with resistance gene analysis, revealed that *E. coli* HD-593 exhibited multidrug resistance. Furthermore, artificial infection experiments and histopathological analysis were performed to evaluate the pathogenicity of *E. coli* HD-593. This investigation not only identified the pathogen responsible for the disease outbreak in *P. sinensis* but also offered valuable data and theoretical support for promoting the rational use of antibiotics in aquaculture.

## 2. Materials and Methods

### 2.1. Animal

The diseased *P. sinensis* (*n* = 11) weighing approximately 1100 ± 30 g were acquired from a breeding facility in Hubei Province, China. These diseased individuals were transported to the laboratory for pathogen isolation. Healthy *P. sinensis* (*n* = 200) with no history of the disease were obtained from the same breeding facility. The healthy *P. sinensis* were temporarily housed in laboratory aquariums measuring 350 cm × 200 cm × 50 cm and maintained at water temperatures of 30 ± 2 °C for 14 days. Healthy *P. sinensis* were randomly divided into six groups, with 30 *P. sinensis* in each group. All animal experiments conducted in this investigation were approved by the Animal Experimental Ethical Inspection of the Laboratory Animal Center, Yangtze River Fisheries Research Institute, Chinese Academy of Fishery Sciences (ID Number: YFI 2023-zhouyong-101).

### 2.2. Pathogen Isolation

Diseased *P. sinensis* was anesthetized, and then disinfected with 75% ethanol. After dissection, some liver tissue was inoculated onto sterile agar plates containing brain heart infusion (BHI) medium (HopeBio, Qingdao, China). These plates were incubated at 30 °C for 24 h. Next, single colonies were selected and purified by repeated streaking on fresh BHI agar plates. The purified bacterial strain was designated HD-593.

### 2.3. Morphological Characteristic

The bacteria were preserved by inoculation into BHI liquid medium and cultured at 30 °C with stirring at 200 rpm for 24 h. The cultured bacterial fluids were diluted for Gram staining (Jiancheng, Nanjing, China), and optical micrographs (Olympus, Tokyo, Japan) were used to visualize bacterial morphology [13]. The bacterium was then stabilized in 2.5% glutaraldehyde solution and dehydrated to dryness [14]. Bacterial morphology was examined using scanning electron microscopy (Hitachi, Tokyo, Japan).

### 2.4. Biochemical Characterization

The biochemical characterization of HD-593 was performed using the Microbial Identification and Phenotype MicroArrays System (MIPAS). The HD-593 strain was introduced into IF-A inoculation liquid (Biolog, Hayward, CA, USA). The inoculum fluid was then dispensed at 100 μL per well onto GEN III recognition plates. The GEN III plate was validated automatically by inserting it into the Biolog Automated Microbiology Recognition System (Biolog, Hayward, CA, USA).

### 2.5. Molecular Biology Analysis

The HD-593 strain was cultured at 30 °C and 200 rpm for 18 h. Subsequently, the cultured bacteria were centrifuged at 12,000 rpm for 10 min. DNA was extracted from the bacterial pellets using the Bacterial DNA Extraction Kit (Tiangen, Beijing, China) to serve as templates for subsequent PCR reactions. The 16S rRNA gene sequence was amplified using the universal primers 27F and 1492R. The reactions contained 12.5 μL 2× Taq PCR Master Mix (Tiangen, Beijing, China), 1 μL of each primer (Huayu Gene, Wuhan, China), 1 μL of template DNA, and deionized water. The amplification program was as follows: 95 °C for 5 min, followed by 35 cycles of 94 °C for 1 min, 55 °C for 1 min, 72 °C for 1 min, and 72 °C for 10 min. The amplified products were analyzed using 1% agarose gel electrophoresis and visualized with an ultraviolet light transilluminator (Bio-Rad, Hercules, CA, USA). These sequences were compared using BLAST on the National Center for Biotechnology Information (NCBI) GenBank database (https://blast.ncbi.nlm.nih.gov/Blast.cgi, accessed on 5 November 2023). The phylogenetic tree was constructed using MEGA software (Version 11.0.13).

### 2.6. Drug Susceptibility and Drug Resistance Genes Testing

The antibiotic sensitization of HD-593 was determined using the Kirby–Bauer disc diffusion technique. The concentration of bacteria was adjusted to 1 × 10^8^ CFU/mL using sterile PBS. Subsequently, 100 μL of the bacterial suspension was uniformly coated on the MHA medium, and antibiotic discs (ceftriaxone, enrofloxacin, doxycycline, sulfonamide, gentamycin, neomycin, florfenicol, carbenicillin, cefradine, erythromycin, penicillin, ampicillin, midecamycin, streptomycin, furazolidone, and polymyxin; diameter: 6 mm) were placed on the plate. After overnight incubation at 30 °C, the diameters of the inhibition zones were measured.

The drug resistance genes–quinolone resistance genes (*oqxA*, *oqxB*), aminoglycoside resistance genes (*aac(3)-II*, *aphA1*), a β-lactam resistance gene (*blaTEM*), an acylaminol resistance gene (*floR*), a colistin resistance gene (*mcr-4*), and a macrolide resistance gene (*ereA*)—were screened using PCR. The primers used for amplification of the drug resistance genes are listed in Appendix A.

### 2.7. Artificial Infection Experiment

The concentrations of the cultured bacterial solution were measured using the plate count method. For infection group, 1 × 10^4^, 1 × 10^5^, 1 × 10^6^, 1 × 10^7^, and 1 × 10^8^ CFU/g body weight bacterial suspension were prepared. The control groups were also injected with an equal volume of phosphate buffer solution (PBS, Procell, Wuhan, China). After infection, the *P. sinensis* were continuously observed for 10 days. Dead *P. sinensis* were subjected to bacterial isolation and characterization. The Reed–Muench method was used to determine the lethal dosage (LD_50_) of HD-593 in *P. sinensis*.

### 2.8. Serum Physiologic and Biochemical Analysis

To investigate the serum alterations in diseased *P. sinensis*, blood samples were collected and transferred to labeled tubes for natural coagulation. Following a 30 min incubation at 4 °C for clot formation, the specimens were centrifuged at 2000 rpm for 10 min to separate the serum and cellular components. The resulting supernatant serum was aliquoted into fresh centrifuge tubes for subsequent biochemical analysis. Serum levels of aspartate aminotransferase (AST), alanine aminotransferase (ALT), alkaline phosphatase (ALP), total protein (TP), albumin (ALB), and globulin (GLB) were evaluated using an automated biochemical analyzer (Sysmex, Kobe, Japan).

### 2.9. Histopathologic Observation

The livers, spleens, kidneys, and intestines from both naturally diseased and healthy *P. sinensis* were collected. These organs were fixed in 4% paraformaldehyde solution, embedded in paraffin, sectioned, and subsequently stained with hematoxylin–eosin (HE). The stained sections were then sealed with neutral resin. The prepared sections were examined under a light microscope (Olympus, Tokyo, Japan) to observe any pathological changes.

## 3. Results

### 3.1. Clinical Symptom and Histopathological Findings

Diseased *P. sinensis* exhibited sluggish movement, staying close to the shore and allowing easy capture. Boil sores were observed on their backs and abdomens (Figure 1A,B). An autopsy revealed significant enlargement of vital organs, including the liver, kidneys, and spleen. The intestines were empty and showed signs of hemorrhaging, suggesting impaired digestion and potential internal bleeding (Figure 1C–E). Diseased *P. sinensis* exhibited multiorgan lesions, including intestinal villous necrosis and detachment with mucosal edema, splenic necrosis and nuclear margination, renal glomerular swelling and tubuloepithelial necrosis with inflammation, and hepatic steatosis with sinusoidal dilation and inflammatory infiltration (Appendix A).

### 3.2. Morphological Features

The isolated HD-593 strain showed distinct growth characteristics on solid culture media. The colonies had a smooth, moist, and creamy white appearance with a characteristic odor (Figure 2A). Gram staining showed Gram-negative characteristics, indicated by red coloration under the microscope (Figure 2B). Scanning electron microscopy (SEM) further confirmed the bar-shaped morphology of the bacteria, with an average length of about 2 μm (Figure 2C).

### 3.3. Physiological and Biochemical Characterization of Bacteria

The HD-593 strain was identified as *Escherichia coli* utilizing the Biolog Microbial Automated Identification System (Biolog, Hayward, CA, USA). Appendix A details the specific metabolic responses and the decision results.

### 3.4. 16S rRNA Gene Sequences Analysis

The 16S rRNA gene of HD-593 is 1374 base pairs (GenBank accession number: PP155069). BLASTn analysis against the NCBI databases (http://blast.ncbi.nlm.nih.gov/, accessed on 5 November 2023) revealed high sequence similarity (>99%) between the HD-593 16S rRNA gene and those of three bacterial strains—KP868689.1, KT260578.1, and MH619508.1—all identified as E. coli. Phylogenetic analysis of the aligned sequences, employing the neighbor-joining method, was then conducted to elucidate the evolutionary relationships between HD-593 and the reference strains. As shown in Figure 3, strain HD-593 shares the same branch of the evolutionary tree as members of the genus *E. coli*.

### 3.5. Drug Sensitivity and Drug Resistance Genes Tests

The drug susceptibility tests conducted on strain HD-593 are summarized in Appendix A. The findings indicated that HD-593 exhibited resistance to ceftriaxone, enrofloxacin, doxycycline, sulfonamide, gentamicin, neomycin, florfenicol, carbenicillin, cefradine, erythromycin, penicillin, ampicillin, medithromycin, and streptomycin. Additionally, the isolate demonstrated moderate susceptibility to polymyxin and furazolidone.

According to the PCR profiles of the drug resistance genes, quinolone resistance genes (*oqxA*, *oqxB*), aminoglycoside resistance genes (*aac(3)-II*, *aphA1*), a β-lactam resistance gene (*blaTEM*), and an acylaminol resistance gene (*floR*) were present; however, the colistin resistance gene (*mcr-4*) and the macrolide resistance gene (*ereA*) were not detected (Figure 4).

### 3.6. Pathogenicity

The infected group exhibited mortality across all tested concentrations; however, there was no death in the control group (Figure 5). The group with the highest mortality rate was the one where *P. sinensis* had a bacterial load of 1 × 10^8^ CFU/g, reaching a mortality rate of 100%. The LD_50_ value of HD-593 was 6.53 × 10^5^ CFU/g *P. sinensis* weight. In the case of artificial infestation, the physical symptoms of the *P. sinensis* mirrored those of the naturally occurring disease, and *E. coli* was re-isolated from the artificially infested deceased *P. sinensis*.

### 3.7. Serum Biochemistry Analysis

According to the serum biochemistry assays (Figure 6), the TP, ALB, and GLB levels in diseased *P. sinensis* were 14.3 g/L, 4.2 g/L, and 10.5 g/L, respectively, showing a marked decrease compared to those in healthy *P. sinensis*. Conversely, the levels of ASP, ALP, and ALT in diseased *P. sinensis* were 320.3 U/L, 468.3 U/L, and 19.6 U/L, respectively, which were significantly elevated compared to the healthy group.

## 4. Discussion

With the rapid development of the aquaculture industry, the need for the use of antibiotics to prevent and treat bacterial diseases is increasing. However, only 20% to 30% of the antibiotics added in aquaculture can be absorbed by the animal body, and the rest are discharged into the environment in the form of raw drugs or metabolites [15]. Antibiotics that remain in animals can cause food safety hazards [16]. Antibiotics that enter the environment are dispersed in water bodies and sediments, so that drug-resistant bacteria and drug-resistant genes in farms and the surrounding environment continue to increase, and even multiple-drug-resistant genes appear [17,18]. Unlike traditional environmental pollutants, drug-resistant genes are a new type of pollutant that is difficult to control. It can be horizontally transmitted to other bacteria through mobile genetic elements such as plasmids, integrons, and transposons, and has the characteristics of replicable transmission and environmental durability [19].

*E. coli* is an important indicator bacteria in food and environmental monitoring [20]. Functioning as a significant zoonotic pathogen, it is an obligate anaerobic Gram-negative bacillus and is now understood to be widely distributed in the intestinal tracts of both humans and animals, as well as prevalent in soil and water environments [21]. While the majority of *E. coli* strains are considered non-pathogenic to humans [22], certain variants can become pathogenic, leading to severe illnesses through intestinal infections [23]. Reports indicate the detection of *E. coli* in various animals, including chickens [24], pigs [25], and sheep [26]. Although there have been limited instances of *E. coli* isolation in aquatic animals [27], the transfer of pathogenic strains can occur in aquatic environments contaminated with *E. coli*. In a prior study, *E. coli* was found to induce intestinal and liver hemorrhage in *crucian carp* [28]. In this study, a strain of *E. coli* was isolated from diseased *P. sinensis*. The primary symptoms observed in the afflicted *P. sinensis* included enlarged liver, kidneys, and spleen, with no evidence of intestinal bleeding due to ingested food. These symptoms closely resembled the outcomes of the present experiment. Morphological characterization, bacterial biochemical identification, and molecular biology confirmed the identification of the isolated bacterium as *E. coli*.

The drug sensitivity tests showed that the isolated *E. coli* strains displayed resistance to ceftriaxone, enrofloxacin, doxycycline, sulfadiazine, gentamicin, neomycin, fosfomycin, carbenicillin, cefradine, erythromycin, penicillin, ampicillin, madicillin, and streptomycin. Additionally, they exhibited moderate susceptibility to furazolidone and polymyxin. Presently, antibiotics serve as the primary means for treating and preventing *E. coli* infections. However, the misuse of antibiotics has led to significant drug resistance issues and diminished efficacy. Furthermore, the *E. coli* HD-593 contained several resistance genes to these drugs, such as quinolone resistance genes (*oqxA*, *oqxB*), aminoglycoside resistance genes (*aac(3)-II*, *aphA1*), a β-lactam resistance gene (*blaTEM*), and an acylaminol resistance gene (*floR*). Moreover, there is a notable concern regarding drug residues, which not only hinders the effective control of *E. coli* disease but may also contribute to the emergence of superbugs, severely impacting the effectiveness of *E. coli* prevention and treatment [26,29].

Regression infection experiments successfully established the pathogenicity of the isolated strains in *P. sinensis*. Previous investigations determined the LD_50_ values of *A. veronii* and *B. cereus* in *P. sinensis* to be 4.17 × 10^5^ CFU/g and 6.80 × 10^3^ CFU/g, respectively [5,7]. In this study, pathogenicity tests revealed that the LD_50_ of *E. coli* on *P. sinensis* was 6.53 × 10^5^ CFU/g, indicating that the pathogenicity of *E. coli* HD-593 on *P. sinensis* was slightly lower than the above three other aquatic pathogens. However, the experimental design focused on acute infection models under controlled laboratory conditions, which may not fully replicate the multifactorial stressors (e.g., water quality fluctuations, co-infections) present in natural aquaculture settings.

Histopathological diagnosis stands as one of the foremost clinical approaches in contemporary medicine [5]. In a previous study, *E. coli* was identified as the causative agent for inflammatory cell infiltration in the liver and spleen of chickens [24]. Additionally, *E. coli* induced the shedding of intestinal villi and inflammatory cell infiltration in the spleen of crucian carp [28]. These findings align with the observations in the present study involving *P. sinensis*, where shedding of intestinal villi, necrosis of splenic cells, and infiltration of inflammatory cells in the liver were evident. The study suggests that *E. coli* infection in *P. sinensis* resulted in severe lesions in various tissues, including the liver, intestines, spleen, and kidneys, likely serving as a primary factor in the mortality of *P. sinensis*.

Animal serum plays a crucial role in their physiological and biochemical adjustments. In the sera of *P. sinensis* infected with *A. veronii*, there was a decrease in the levels of TP, ALB, and GLB. Meanwhile, the levels of AST, ALP, and ALT were elevated compared to non-infected *P. sinensis* [7]. In our study, a significant reduction in the proportions of serum biochemical molecules TP, ALB, and GLB was observed, indicating liver impairment, disease, and a weakened immune system. Concurrently, AST, ALP, and ALT levels were markedly elevated, suggesting inflammation in the liver and myocardium. These findings align with the results of the current experiment.

## 5. Conclusions

This study demonstrates that multidrug-resistant *E. coli* HD-593, isolated from diseased *P. sinensis*, poses a significant threat to aquaculture health and environmental safety. The strain exhibited resistance to 14 antibiotics and harbored critical resistance genes, highlighting risks of horizontal gene transfer and persistent environmental contamination. Pathogenicity testing confirmed its capacity to induce systemic infections in *P. sinensis*, with histopathological and serum biochemical analyses revealing severe multiorgan damage. These findings underscore the urgent need for antibiotic stewardship in aquaculture to mitigate the emergence of resistant pathogens. Future work should prioritize surveillance of resistance gene dissemination and explore alternatives to antibiotics, such as vaccines or probiotics, to safeguard both animal and public health.

## Figures and Tables

**Figure 1 vetsci-12-00473-f001:**
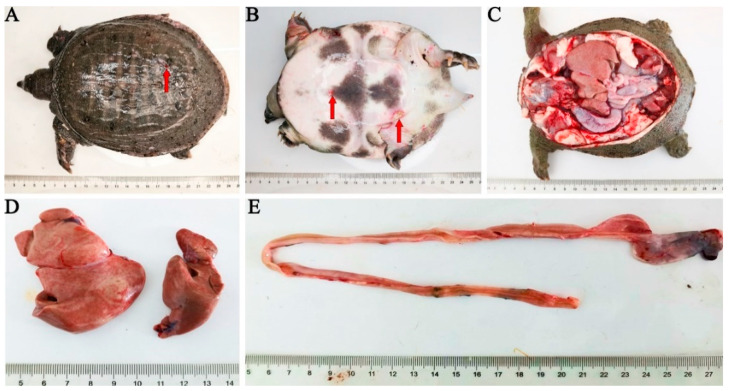
Presentation of the diseased *P. sinensis*. (**A**): Boils on the backside (red arrows), (**B**): boils on the abdomen (red arrows), (**C**): anatomy of sick turtle, (**D**): enlargement of the liver, (**E**): no food and bleeding in the intestines.

**Figure 2 vetsci-12-00473-f002:**
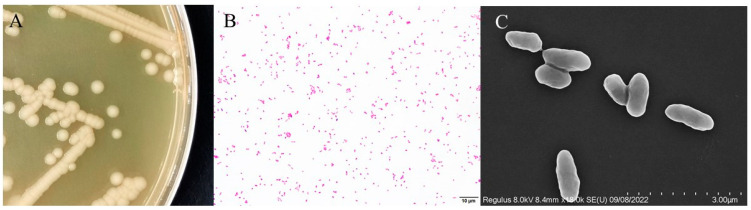
Morphological features of HD-593 bacteria. (**A**): HD-593 bacteria growing in BHI solid medium. (**B**): HD-593 bacteria Gram stain (scale bar: 10 μm). (**C**): The scanning electronic micrograph of the HD-593 (scale bar: 3 μm).

**Figure 3 vetsci-12-00473-f003:**
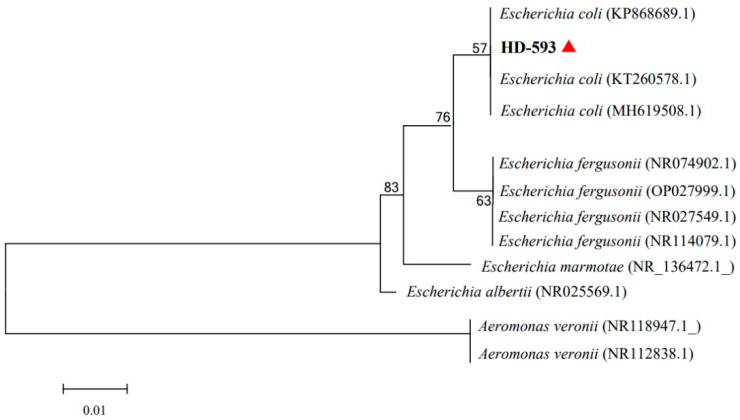
The phylogenetic tree of the HD-593 (red triangle) isolate based on the neighbor-joining technique in MEGA 11. Each branch is annotated with a percentage, indicating the bootstrap values based on 1000 replicates. The scale bars represent the number of substitutions per locus.

**Figure 4 vetsci-12-00473-f004:**
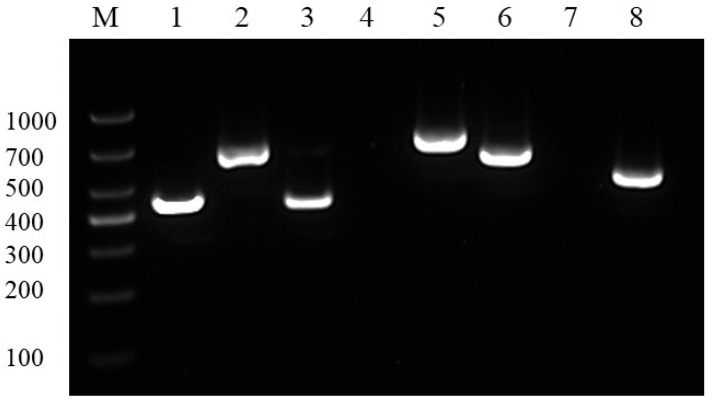
Agarose gel electrophoresis of PCR products of the drug resistance genes. M: DL1000 marker, 1: *blaTEM*, 2: *oqxA*, 3: *aac(3)-II*, 4: *mcr-4*, 5: *floR*, 6: *aphA1*, 7: *ereA*, 8: *oqxB*. Original image—Appendix A.

**Figure 5 vetsci-12-00473-f005:**
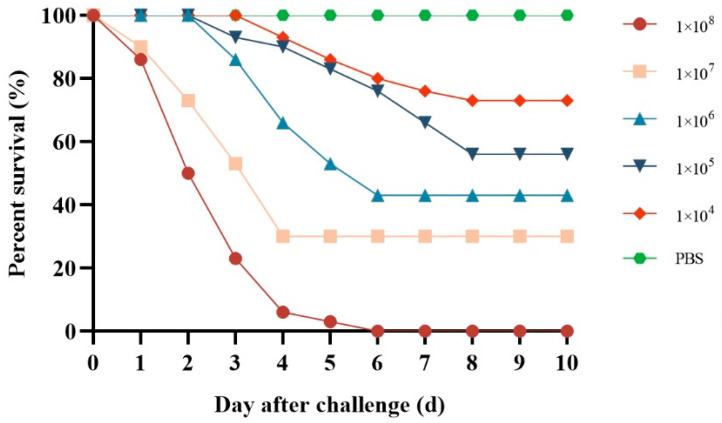
Cumulative mortality of *P. sinensis* infected with *E. coli* HD-593 at different bacterial doses (1 × 10^4^–1 × 10^8^ CFU/g body weight). Groups (n = 30/group) were intraperitoneally injected with bacterial suspensions or PBS (control). Mortality was monitored for 10 days. Mortality reached 100% at the highest dose (1 × 10^8^ CFU/g). The LD_50_ value (6.53 × 10^5^ CFU/g) was calculated using the Reed–Muench method.

**Figure 6 vetsci-12-00473-f006:**
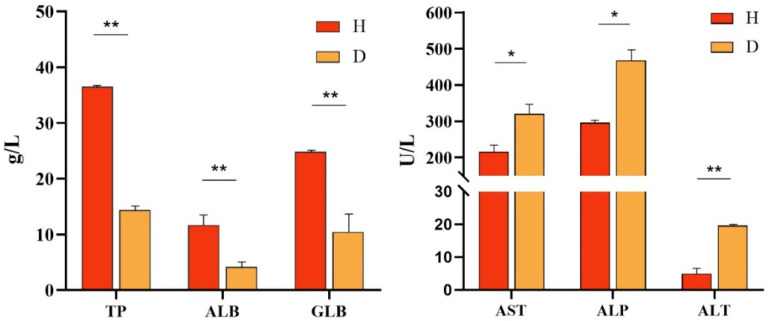
Serum biochemical parameters in healthy (H) and diseased (D) *P. sinensis*. Levels of total protein (TP), albumin (ALB), globulin (GLB), aspartate aminotransferase (AST), alkaline phosphatase (ALP), and alanine aminotransferase (ALT) were quantified. Values are mean ± SD (* *p* < 0.05; ** *p* < 0.01).

## Data Availability

The data are contained within this paper and its Appendix A.

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
