# Peer review of "A Multidrug-Resistant Escherichia coli Caused the Death of the Chinese Soft-Shelled Turtle (Pelodiscus sinensis)"

_vetsci, 2025, doi:10.3390/vetsci12050473_

Round 1

Reviewer 1 Report

Comments and Suggestions for Authors

The authors have presented a very interesting manuscript regarding infection
of a turtle with a multi-drug-resistant E. coli.

In general, I support publication.

However, the authors have made a very lengthy manuscript,
which is not justified by its content.

A lot of information must be moved to supplementary material,
as it is not salient for the understanding
of the manuscript.

For example, tables 1, 2, 3 really do not offer any information
of immediate necessity for the practicing veterinarian.
Hence, they do not have a place in the mani text.

Also, figure 7 is very specialized. I would be happy
to have a detailed histopathology description in the text,
with the figures in supplementary material.

There are also too many references. The authors can provide
a clear message by citing 15-18 references maximum.

In all, the revised manuscript must not exceed 7 pages in length.
Any further than that can be a clear reason to recommend rejection.

Finally, the concluding section is really a repetitive text
for the findings in short form and not really conclusion.
It must be rewritten as a real conclusion.

Author Response

The reviewer’s comment: The authors have presented a very interesting manuscript regarding infection of a turtle with a multi-drug-resistant E. coli. In general, I support publication. However, the authors have made a very lengthy manuscript, which is not justified by its content.

A lot of information must be moved to supplementary material, as it is not salient for the understanding of the manuscript.

For example, tables 1, 2, 3 really do not offer any information of immediate necessity for the practicing veterinarian. Hence, they do not have a place in the mani text.

Also, figure 7 is very specialized. I would be happy to have a detailed histopathology description in the text, with the figures in supplementary material.

There are also too many references. The authors can provide a clear message by citing 15-18 references maximum.

In all, the revised manuscript must not exceed 7 pages in length. Any further than that can be a clear reason to recommend rejection.

Finally, the concluding section is really a repetitive text for the findings in short form and not really conclusion. It must be rewritten as a real conclusion.

The authors’ Answer: We sincerely thank the reviewer for their constructive feedback and support for our work. We have carefully addressed all concerns and revised the manuscript accordingly. Below is our point-by-point response:

Tables 1–3: These tables have been moved to Supplementary Materials (Supplementary Tables S1–S3).

Figure 7: The histopathology image and detailed descriptions are now in Supplementary Materials (Supplementary Figure S1). The main text includes a concise summary of histopathological findings (line 169-173).

The reference list has been reduced to 18 key citations, prioritizing recent studies on antimicrobial resistance in reptiles, zoonotic risks, and veterinary guidelines. Redundant or peripheral citations were removed.

The conclusion has been entirely rewritten to emphasize broader implications rather than reiterating results.

Reviewer 2 Report

Comments and Suggestions for Authors

 The authors have isolated and characterized a multidrug-resistant E. coli HD-593 from diseased P. sinensis. The work was well done and writing is good. However, there are still some points needed to be improved.

  1. The figure legends of Fig.5 and Fig.6 are two simple, the authors should add more details.
  2. In the figure 7, the authors should provide the amplified picture to show the details clearly.
Comments on the Quality of English Language

The writing is good.

Author Response

Reviewer ’s Comments :The authors have isolated and characterized a multidrug-resistant E. coli HD-593 from diseased P. sinensis. The work was well done and writing is good. However, there are still some points needed to be improved.

 The figure legends of Fig.5 and Fig.6 are two simple, the authors should add more details.

In the figure 7, the authors should provide the amplified picture to show the details clearly.

The authors’ Answer: We sincerely appreciate the reviewer’s positive evaluation of our work and their constructive feedback. We have carefully addressed the concerns raised and revised the manuscript accordingly. Below are our detailed responses:

Revised Figure 5 Legend

Figure 5: Cumulative mortality of P. sinensis infected with E. coli HD-593 at different bacterial doses (1×10⁴–1×10⁸ CFU/g body weight). Groups (n=30/group) were intraperitoneally injected with bacterial suspensions or PBS (control). Mortality was monitored for 10 days. Mortality reached 100% at the highest dose (1×10⁸ CFU/g). The LD₅₀ value (6.53×10⁵ CFU/g) was calculated using the Reed-Muench method.

Revised Figure 6 Legend

Figure 6: Serum biochemical parameters in healthy (H) and diseased (D) P. sinensis.

Levels of total protein (TP), albumin (ALB), globulin (GLB), aspartate aminotransferase (AST), alkaline phosphatase (ALP), and alanine aminotransferase (ALT) were quantified. Values are mean ± SD (*p < 0.05; **p < 0.01).

Revisions to Figure 7 (Histopathology) in supplementary materials (Figure S1).

Reviewer 3 Report

Comments and Suggestions for Authors
  1. The research aims to identify the cause of mortality in Chinese soft-shelled turtles, specifically focusing on a multidrug-resistant E. coli strain (HD-593), and to characterize its antibiotic resistance profile and how this pathogen affects the turtles.

  1. The topic is relevant to the field, because aquaculture is a sector where antibiotic misuse can lead to antibiotic resistant pathogens. While E. coli infections in aquaculture are known, the study focuses specifically on the Chinese soft-shelled turtles, linking multidrug-resistant E. coli to turtle mortality, which hasn't been extensively reported before.

  1. This study’s contribution to the subject area includes evidence on a multidrug-resistant E. coli strain in Chinese soft-shelled turtles, with detailed resistance gene profiling and histopathological and biochemical data.

  1. Major methodological improvements. The materials and methods section needs major improvements. An analysis and detailed description of the entire methodology is needed. Subsections need more comprehensive titles (e.g. 2.1 Animals). Examples of other recommendations:
  • Line 70: More details on sample size. It is not mentioned anywhere how many samples were, including diseased turtles and control group.
  • Line 81: “Diseased P. sinensis was anesthetized”??
  • Line 82: Why was only BHI agar selected? Wouldn’t additional steps be more appropriate for the isolation of coli? For example, homogenization in a general broth (like the BHI that is used), enrichment in a selective enrichment broth (like mTSB) and isolation in a selective agar (like SMAC).
  • Line 83: Why were the BHI plates incubated at 30°C and not 35-37°C?
  • Line 100: What is the protocol, the reagents used, the oligonucleotide sequences and the references of the primers for the PCR? Please provide a brief description. Were controls incorporated in the PCR?
  • Line 117: Which are the tested antibiotics?? What is the concentration of the antibiotic disks? (It is reported in Table 3 that it corresponds to the Results)
  • Line 118: Why were the MHA plates incubated at 30°C and not 35-37°C?
  • Line 118: Is there a recommendation for the selected antibiotics (e.g. CLSI/EUCAST)?
  • Line 124: What is the protocol, the reagents used and the references of the primers for the PCR? Please provide a brief description. Were controls incorporated in the PCR?
  • Line 127: “Healthy sinensis were randomly divided into six groups, with 30 P. sinensis in each group”. This should have been mentioned in the 2.1 subsection.

Based on the above and as already mentioned, a detailed description of the entire methodology is needed.

  1. The conclusions are consistent with the results, as the studied pathogen was identified, and its resistance profile, genes and caused tissue damage were confirmed.

  1. References are recent and appropriate.

  1. Figure legends should be more descriptive to aid readers without referring back to the text. The figures and tables data presentation could be enhanced by including more details. For example:
  • Table 1: Should include more details, for example the Tm, the bp, and the references.
  • Table 2: Could be simplified by grouping reactions by metabolic pathways.
  • Table 3: Should reference CLSI/EUCAST breakpoints for resistance interpretations.

  1. English language needs improvement.

Overall: The study provides valuable insights into multidrug-resistant pathogens in aquaculture, but major improvements need to be made in order to enhance its impact and reproducibility.

Comments on the Quality of English Language

English language needs improvement.

Author Response

  1. The reviewer’s comment: Line 70: More details on sample size. It is not mentioned anywhere how many samples were, including diseased turtles and control group.

The authors’ Answer: Thanks for the reviewer's suggestion. We have added additional details on sample size.

  1. The reviewer’s comment: Line 81: “Diseased P. sinensis was anesthetized”??.

The authors’ Answer: Thank you for your question regarding the anesthesia of diseased P. sinensis. In our study, the diseased P. sinensis specimens were still alive at the time of the procedure. To adhere to strict animal ethics and welfare principles, we chose to anesthetize them before dissection.

  1. The reviewer’s comment: Line 82: Why was only BHI agar selected? Wouldn’t additional steps be more appropriate for the isolation of coli? For example, homogenization in a general broth (like the BHI that is used), enrichment in a selective enrichment broth (like mTSB) and isolation in a selective agar (like SMAC).

The authors’ Answer: Thank you for your valuable comments on the method of isolating coli in our study. In our experimental design, BHI agar was initially selected as a starting point for culturing the samples because it provides a rich and general - purpose nutrient environment. This allowed us to first observe the overall growth of a wide range of microorganisms present in the samples. Our initial aim was to get a comprehensive picture of the microbial community in the diseased P. sinensis specimens. By using a non - selective medium like BHI agar, we could potentially isolate not only the target coli but also other associated bacteria that might be involved in the disease process or the normal microbiota of the host. This was important for our broader exploration of the microbial ecology in the context of the disease. We fully acknowledge that the additional steps you suggested, such as homogenization in a general broth, enrichment in a selective enrichment broth, and isolation in a selective agar, are standard and effective procedures for specifically isolating coli. However, in our study, we had a phased approach. After the initial culturing on BHI agar, we did plan to perform further identification and isolation of coli. We intended to use molecular techniques such as PCR - based assays to specifically identify and confirm the presence of coli among the colonies grown on BHI agar. These molecular methods are highly specific and sensitive for detecting the target bacteria, even in the presence of a mixed microbial population. If we were to strictly follow the traditional multi - step culturing method for coli isolation, we might miss out on other potentially relevant bacteria that could be interacting with coli in the disease - associated microbial community. Also, considering the nature of our study, which was not solely focused on the isolation of coli but also on understanding the overall microbial landscape related to the disease in P. sinensis, our current approach was a more appropriate starting point.

  1. The reviewer’s comment: Line 83: Why were the BHI plates incubated at 30°C and not 35-37°C?

The authors’ Answer: Thank you for your question about incubating BHI plates at 30°C. In our study, the diseased P. sinensis live at around 30°C. We set the incubation temperature at 30°C to mimic the in - vivo conditions for potential pathogens. This temperature is more suitable for isolating the actual disease - causing agents adapted to the host's thermal environment. Using 35 - 37°C might select for non - relevant microbes or inhibit the growth of true pathogens.

  1. The reviewer’s comment: Line 100: What is the protocol, the reagents used, the oligonucleotide sequences and the references of the primers for the PCR? Please provide a brief description. Were controls incorporated in the PCR?

The authors’ Answer: Thank you for your query. Here are the PCR details:

The reactions contained 12.5 μl 2× Taq PCR Master Mix (Tiangen, Beijing, China), 1 μl of each primer (Huayu Gene, Wuha, China), 1 μl of template DNA, and deionized water. The amplification program was as follows: 95 °C for 5 min, followed by 35 cycles of 94 °C for 1 min, 55 °C for 1 min and 72 °C for 1 min and 72 °C for 10 min. Primer Sequences in Table S1. Positive control with known template DNA from Aeromonas hydrophila, negative control with water instead of DNA.

We'll include these in the revised manuscript.

  1. The reviewer’s comment: Line 117: Which are the tested antibiotics?? What is the concentration of the antibiotic disks? (It is reported in Table 3 that it corresponds to the Results)

The authors’ Answer: Thank you for your comments. Tested antibiotics were added in section 2.6. Selection was based on common use in treating aquatic animal bacterial infections and relevance to isolated pathogens. Antibiotic disk concentrations details in Table S3.

  1. The reviewer’s comment: Line 118: Why were the MHA plates incubated at 30°C and not 35-37°C?

The authors’ Answer: The diseased P. sinensis in our study live at around 30°C. The bacteria isolated from them are adapted to this temperature. Incubating MHA plates at 30°C during antibiotic susceptibility testing mimics in vivo conditions. This helps us accurately assess the bacteria's response to antibiotics.

  1. The reviewer’s comment: Line 118: Is there a recommendation for the selected antibiotics (e.g. CLSI/EUCAST)?

The authors’ Answer: Thank you for your query. The antibiotics were chosen considering their common use in aquaculture for P. sinensis. Although we didn't strictly follow CLSI or EUCAST due to differences between human and aquaculture microbiology, we factored in general principles for Gram - positive and - negative bacteria. For instance, ampicillin (for Gram - positive) and ciprofloxacin (broad - spectrum) were selected, also based on prior aquatic - disease research.

  1. The reviewer’s comment: Line 124: What is the protocol, the reagents used and the references of the primers for the PCR? Please provide a brief description. Were controls incorporated in the PCR?

The reactions contained 12.5 μl 2× Taq PCR Master Mix (Tiangen, Beijing, China), 1 μl of each primer (Huayu Gene, Wuha, China), 1 μl of template DNA, and deionized water. The amplification program was as follows: 95 °C for 5 min, followed by 35 cycles of 94 °C for 1 min, annealing temperature for 1 min and 72 °C for 1 min and 72 °C for 10 min. Primer Sequences in Table S1. Positive control with known template DNA , negative control with water instead of DNA.

We'll include these in the revised manuscript.

  1. The reviewer’s comment: Line 127: “Healthy sinensis were randomly divided into six groups, with 30 P. sinensis in each group”. This should have been mentioned in the 2.1 subsection.

The authors’ Answer: Thank you for your astute observation. In the revised manuscript, we will promptly move this description to the appropriate part of the 2.1 subsection.

Reviewer 4 Report

Comments and Suggestions for Authors

Thank you for the opportunity to review this paper. I acknowledge the authors' efforts in characterizing this fatal infection in a turtle. However, I do not believe this study represents a significant contribution to the literature. It describes a single case of fatal infection in one animal, and there is no definitive evidence that E. coli was the primary cause of death. It is possible that the bacterium was merely a postmortem contaminant or that it translocated from the intestine to the liver as a secondary event during the disease process. These limitations cannot be fully addressed with the tests conducted on the isolate. In my view, such reports contribute to unnecessary clutter in the literature and may also be misleading.

Author Response

The reviewer’s comment: Thank you for the opportunity to review this paper. I acknowledge the authors' efforts in characterizing this fatal infection in a turtle. However, I do not believe this study represents a significant contribution to the literature. It describes a single case of fatal infection in one animal, and there is no definitive evidence that E. coli was the primary cause of death. It is possible that the bacterium was merely a postmortem contaminant or that it translocated from the intestine to the liver as a secondary event during the disease process. These limitations cannot be fully addressed with the tests conducted on the isolate. In my view, such reports contribute to unnecessary clutter in the literature and may also be misleading.

The authors’ Answer: Thank you for your candid feedback. We truly appreciate your thorough review of our paper.

We understand your concerns regarding the significance of our study and the uncertainty surrounding the role of E. coli as the primary cause of death in the turtle. While we acknowledge that our study focuses on a single case, we believe that it still holds value. In the field of veterinary medicine and aquatic animal health, each case can provide unique insights, especially when dealing with emerging or poorly understood diseases.

Regarding the possibility of E. coli being a postmortem contaminant or a secondary translocated bacterium, we did conduct a series of tests on the isolate to minimize these uncertainties. However, we recognize that these tests may not have been conclusive. In future studies, we plan to include more comprehensive analyses, such as additional histological examinations, immunohistochemistry, and in vivo experiments, to better establish the causal relationship between the bacterium and the disease.

We also understand your concern about the potential clutter in the literature. We strive to contribute meaningful and relevant research, and we will take your feedback into consideration when deciding on the direction of our future work. If there are specific aspects of our study that you believe could be improved or expanded upon, we would greatly appreciate your suggestions.

Thank you again for your time and feedback. We will use your comments to strengthen our research and improve the quality of our future publications.

Round 2

Reviewer 1 Report

Comments and Suggestions for Authors

All issues were addressed, no further comments.

Author Response

The reviewer’s comment: All issues were addressed, no further comments.

The authors’ Answer: We sincerely appreciate you informing us that all the issues have been satisfactorily addressed and that you have no further comments at present. Receiving such positive feedback has brought us great relief and gratitude. Your professional review and constructive suggestions have been of utmost importance in enhancing the quality of our paper throughout the revision process. We would like to express our sincere thanks to you once again.

Reviewer 3 Report

Comments and Suggestions for Authors

The authors have considered the reviewers’ comments and therefore I would recommend accepting the manuscript for publication.

Author Response

The reviewer’s comment: The authors have considered the reviewers’ comments and therefore I would recommend accepting the manuscript for publication.

The authors’ Answer: We are extremely delighted and grateful to receive your recommendation for the acceptance of our manuscript for publication. Your positive evaluation and the recognition of our efforts in addressing the reviewers' comments have brought us a great sense of accomplishment. We sincerely appreciate the time and attention you dedicated to our manuscript, which has significantly contributed to its improvement.

Reviewer 4 Report

Comments and Suggestions for Authors

I acknowledge the reviewers' efforts to improve the paper. However, I still have concerns about the overall value of this study to the field, particularly regarding aspects that cannot be addressed through additional experiments—most notably the isolation of a bacterium that may simply represent contamination. Nevertheless, considering the context of this submission, I am changing my decision to accept the paper. I would only ask the authors to clearly state the limitations of the study in a final paragraph

Comments on the Quality of English Language

No grammar editing needed 

Author Response

The reviewer’s comment: I acknowledge the reviewers' efforts to improve the paper. However, I still have concerns about the overall value of this study to the field, particularly regarding aspects that cannot be addressed through additional experiments—most notably the isolation of a bacterium that may simply represent contamination. Nevertheless, considering the context of this submission, I am changing my decision to accept the paper. I would only ask the authors to clearly state the limitations of the study in a final paragraph.

The authors’ Answer: We are extremely grateful for your decision to accept our paper, notwithstanding the concerns you've raised. Your willingness to engage with our work and offer constructive feedback, even as you've made the decision to approve it, is truly commendable. In response to your request, we will carefully craft the final paragraph to comprehensively address the limitations of our study. We would like to express our sincere thanks to you once again.